# Origin of Negative Photoconductivity at the Interface of Ba_0.8_Sr_0.2_TiO_3_/LaMnO_3_/Ba_0.8_Sr_0.2_TiO_3_ Heterostructures

**DOI:** 10.3390/nano12213774

**Published:** 2022-10-26

**Authors:** Aleksei O. Chibirev, Andrei V. Leontyev, Viktor V. Kabanov, Rinat F. Mamin

**Affiliations:** 1Zavoisky Physical-Technical Institute, FRC Kazan Scientific Center of RAS, 420029 Kazan, Russia; 2Department for Complex Matter, Jozef Stefan Institute, 1000 Ljubljana, Slovenia

**Keywords:** ferroelectric heterostructures, photoconductivity, thin films, screening

## Abstract

The study of Ba_0.8_Sr_0.2_TiO_3_/LaMnO_3_/Ba_0.8_Sr_0.2_TiO_3_ heterostructures on a MgO substrate with Ba_0.8_Sr_0.2_TiO_3_ ferroelectric films revealed the occurrence of a metallic character of the temperature behavior of the resistance at a temperature less than 175 K. This behavior is associated with an increased charge concentration at the interface due to a discontinuity in the ferroelectric polarization at the interface between the films. At these temperatures, the effect of negative photoconductivity is observed under uniform illumination with the light of a selected spectral composition event on the surface of the ferroelectric film. The combined exposure to green and infrared light led to an addition of the effects. As a result, a cumulative effect was observed. The effect of metallic conductivity is due to the discontinuity of ferroelectric polarization. Therefore, we explain that the partial screening of the ferroelectric polarization by photogenerated charge carriers causes a reduction in the carrier concentration at the interface. Measurements in the Kelvin mode of atomic force microscopy showed that illumination influences the surface charge concentration in a similar way; this observation confirms our hypothesis.

## 1. Introduction

The use of various oxides leads to progress in the development of modern electronic devices and new technologies. In this regard, we should specifically mention the oxides of rare earth elements which have been attracting attention for several decades, enabling the development of technologies in optical applications [1,2,3,4]. Promising fields of application for these oxides with various doping are laser technologies, optoelectronics, and fiber optic applications [1], which, in particular, are used in the development of modern quantum computers [4]. They are also widely used as sensors, detectors, and other electronic devices [2], as well as in sustainable nuclear, radiation, and energy devices [3]. However, creating complex heterostructures can lead to entirely new functional properties with the characteristics of the heterostructured material changing dramatically when one of its parts is acted upon. One apt example is when two non-conducting oxides form a heterostructure that has unique transport properties of high conductivity at the interface region. Two-dimensional electron gas (2DEG) of high mobility was first discovered at the interface between two non-conducting oxides, LaAlO_3_ (LAO) and SrTiO_3_ (STO) [5]. Over the past two decades, these types of heterostructures have been studied extensively [5,6,7,8,9,10,11,12,13]. It was found that metallic conductivity occurs in STO layers at the LaAlO_3_/SrTiO_3_ (LAO/STO) interface when the number of LAO layers becomes greater than three [5,6]. Furthermore, LAO/STO heterostructures were also found to be ferromagnetic [6] and superconducting [7,8,10], with the transition to the superconducting state being observed below 300 mK. Superconducting behavior was also later discovered at temperatures as high as 30 K in Ba_0.8_Sr_0.2_TiO_3_/La_2_CuO_4_ heterostructures [14]. Subsequently, 2DEG has been found in heterostructures consisting of other non-magnetic dielectrics. The factor required for the occurrence of 2DEG is the polarity of one of the components of a heterostructure; in particular, it was shown that states with high conductivity can be observed at the interface between dielectric oxides, with one of those being ferroelectric [15,16]. Recently, two-dimensional superconductivity tunable by an electric field was observed in EuO/KTaO_3_ heterostructures [17].

The effect of negative photoconductivity is observed in different systems [18,19,20], however, it is usually an effect occurring in a single compound over some uniform volume. A different situation occurs in a multi-component system. In this case, the exposure to light of one of the components of the system can lead to a significant change in the conductive properties in another part of the system. As an example of such a multicomponent system, a ferroelectric/dielectric heterostructure with metallic conductivity at the interface may be indicated. The ferroelectric layers in dielectric/ferroelectric/dielectric heterostructures contain polarization domains that can lead to negative capacitance [21], important for manufacturing field-effect transistors [21,22]. Metallic conductivity at the ferroelectric/dielectric interface is also important in the development of new types of field-effect transistors [22]. If the occurrence of a highly conductive region at the interface depends on the polarization of the ferroelectric oxide part, switching the direction of this polarization would allow for commanding the conductive properties of the material. Previously, the transition to the state with 2DEG at the interface, which is connected to a sharp change in the perpendicular component of the polarization at the interface, was demonstrated [22,23,24], and we detected the presence of a highly conductive state at the interface of a heterostructure based on a ferroelectric and a dielectric, Ba_0.8_Sr_0.2_TiO_3_/LaMnO_3_ [22]. While optical, photoconductive, and other properties of rare earth oxides are well known [1,2,3,4], the optical and especially the photoconductive properties of the 2DEG on the interfaces of two insulating materials are not reported. Therefore, in this work, we study the conductivity of a heterostructure consisting of Ba_0.8_Sr_0.2_TiO_3_/LaMnO_3_/Ba_0.8_Sr_0.2_TiO_3_ films deposited on a MgO substrate and the effect of illumination on the conductivity at the interface.

## 2. Experimental Method

The studies were performed on heterostructures consisting of manganate LaMnO_3_ (LMO) and thick ferroelectric Ba_0.8_Sr_0.2_TiO_3_ (BSTO) films on MgO substrate in (100) orientation fabricated by reactive sputtering of stoichiometric targets using the RF-sputtering method [25,26,27] at 650 °C. The thicknesses of the layers were 383 nm, 553 nm, and 453 nm (see Figure 1a,e). The typical topography of the surface of BSTO film of a virgin heterostructure was measured using an atomic force microscope (AFM) in Figure 1b. The average roughness of the virgin heterostructures was about 0.5–1 nm, which is much smaller than the thickness of the top layer. The ferroelectric transition temperature in films in this thickness range should be higher than that in a bulk sample [26,27]. The BSTO film had a ferroelectric transition temperature of about 540 K for 300-nm thick films on MgO substrate [26,27], and films with thicknesses of 383 to 553 nm were used. Therefore, the BSTO film will be in a ferroelectric state below room temperature. The piezoelectric response and the surface potential of the top Ba_0.8_Sr_0.2_TiO_3_ ferroelectric film of the heterostructure were studied by AFM-based piezoresponse and Kelvin probe measurements at room temperature using the commercial scanning probe microscope (Smena-A, NT-MDT, Russia). Figure 1c,d shows the amplitude and the phase of the vertical component of the piezoelectric response on the surface of BSTO film measured by AFM at room temperature. The presence of a piezoelectric response provides evidence in favor of the existence of ferroelectric polarization in the BSTO film. Moreover, the results of X-ray measurements indicate that the BSTO film is in the tetragonal phase, which means that it is in the ferroelectric state. The X-ray diffraction pattern of the heterostructure is shown in Figure 1f (c = 0.4048 nm for BSTO film). The results of the X-ray and piezoresponse measurements show that the polarization is directed predominantly in the vertical direction.

In order to measure the electrical properties of the interface, four fine gauge gold wire leads were attached to the sides of a sample using silver paste (see Figure 1e). Outer electrodes sourced a current (~10–40 µA), and the resistance was calculated from the potential difference measured across the inner pair. A typical electrode layout can be seen in the inset of Figure 2 (the electrodes had been manually re-applied multiple times between the experiments, introducing some variation in electrode placements). The sample was mounted onto the cold finger of a cryostat (continuous flow ST-100, Janis). As a current source, a standard lab power supply with a current limiting resistor was used, and the measurements were conducted using Mastech DC voltmeters connected to a PC, allowing data collection with a 0.5 s resolution. For illumination, a femtosecond laser system was used, based on a Yb-doped fiber oscillator equipped with a Yb-doped diode-pumped solid-state regenerative amplifier (Avesta, Moscow, Russia) generating 200 fs infrared (IR) pulses (1028 nm, 1.2 eV) with a 3 kHz repetition rate. A second harmonics unit was used to produce green (514 nm, 2.4 eV) radiation. In most experiments, the intensity was tuned to 5 µJ per pulse, or 15 mW if averaged over time. This was to avoid temperature instabilities due to laser heating (The long-term heating for these conditions is estimated to be less than 1 K [28,29,30,31,32,33,34,35,36,37]. Some transient heating from ultrashort laser pulses is expected but it dissipates almost instantly [38]). For the purposes of this study, the train of ultrashort pulses was further viewed as quasi-CW radiation, as the detection system wouldn’t allow temporal resolution beyond 0.5 s. Spatially, the illumination beams were quasi-Gaussian, with a waist radius of ~3 mm, and directed at the sample, unfocused, so the area between the inner voltage electrodes was illuminated fully. Both beams could be switched on or off independently by manually controlled shutters.

## 3. Results

Figure 2 shows the typical temperature dependencies of the electrical resistance of the BSTO/LMO/BSTO heterostructure sample in a dark state (without any illumination). It is evident that the resistance decreases significantly when the temperature decreases below 175 K, exhibiting metallic-like behavior. The nature of the state with a high conductivity at the boundary is quite similar to the nature of the conductivity at the charged domain walls in ferroelectrics [39,40,41], but the state with a high concentration of free carriers, in this case, arises at the boundary rather than on the domain walls.

The right shoulder of the curve with the opposite behavior exists because the conductive path of the interface is being bypassed by the conductivity of the LMO film at higher temperatures. Note that the bell-shaped curve was observed only when reliable contact with the interface was provided, typically when using larger contact spots (inset in Figure 2). Otherwise, in cases when the wires were attached to the BSTO layer only (the top of the heterostructure) without sufficient contact with the interface area, metallic conductivity was absent. Moreover, the current through the BSTO layer at lower temperatures was too low to detect.

In the subsequent experiments, the sample temperature was stabilized at one of the selected points within the left shoulder of the resistance-temperature curve to ascertain that the changes in heterointerface conductivity would be noticeable in the measured signal. The sample temperature was kept steady and the electrical resistance was monitored while the area between the voltage electrodes was exposed to light.

The illumination at both green (514 nm) and infrared (1028 nm) wavelengths had a similar effect on the electrical resistance, it, rather notably, increased every time when illuminated, reverting to an initial state in the dark, i.e., a negative photoconductivity effect was observed. The scope of the effect did, however, differ. Figure 3a,b shows the surges in resistance following the repeated exposure to light at different wavelengths, but at an approximately constant intensity, alternating with dark periods. In Figure 3c the intensities were adjusted to provide the same photon flow rate between the green and IR wavelengths. The switching between the higher and lower resistance states occurs on a timescale of 10–50 s with no noticeable dependence on temperature, light intensity, or wavelength. To further ascertain that this time does not reflect a slow temperature drift due to heat from a laser pulse possibly not fully dissipating during the interval before the next pulse, and building up over time, we have tried the following modification of the illumination conditions. If we change the pulse repetition rate, while keeping the average illumination dose constant (Figure 3d), and the accumulated laser heating is non-negligible, the observed switching times would extend by lowering the pulse repetition rate. However, as seen in Figure 3d, this is not the case.

The effect of the simultaneous illumination of a BSTO/LMO/BSTO sample by IR and green light was also studied. In Figure 3e,f, the electrical resistance over time is depicted when the sample is illuminated in a specific sequence. First, while the green light was on, a certain duration of IR light was superimposed. Likewise, under the IR light, the green was turned on and then off. The intensities of green and IR light were identical. We can see that the effects of exposure are additive, with green light contributing more comparatively to IR, and a cumulative effect is observed. 

In order to explain the results, we propose the following structure of the conduction and valence bands and trapping levels (M) for BSTO film as it is shown in Figure 4. Here *n*_o_, *m*_o_, and *p*_o_ are equilibrium concentrations of conduction electrons and electrons at trapping levels and holes in the valence band, N*_c_*, M, and N*_v_* are the densities of states in the conduction band, at trapping levels and in the valence band. The processes of irradiation with green (G) and infrared (R) light are shown schematically. The processes of relaxation from the conduction band to the valence band (1) and to trapping levels (3) and from the trapping levels to the conduction band (2), and also the thermal activation processes from the trapping levels to the conduction band (4) and from the valence band to the trapping levels (5) are presented as well. Based on the energy level structure shown in Figure 4, illumination with green light populates trapping centers, followed by thermal activation to the conduction band (activation energy *u* ≈ 0.6–0.8 eV, the gap between the conduction and the valence bands *E*_g_ ≈ 3.0–3.4 eV [42]). In this case, free current carriers are created both in the conduction band and in the valence band. Infrared light directly excites trapped carriers into the conduction band. The difference with green illumination is that free current carriers are created in the conduction band only. It should be noted that we made an estimate of the heating within the simple stationary model of thermal diffusion. The estimate shows that with our parameters, the excess temperature Δ*T* is less than 0.1 K. Our estimates are consistent with the estimates of Δ*T* = 12 K which were performed for the focused light, at a much higher intensity and with strong absorption in the surface region [32]. Therefore, we believe that the heating due to the laser beam is not enough to explain the effect of negative photoconductivity.

Let us discuss now how this charge transfer mechanism can influence the resistance following photostimulation. Negative photoconductivity in BSTO due to, i.e., the photogeneration of trap states, has not been observed and would not be relevant since the BSTO film conductivity was deemed to be too low to detect in these experiments in the first place. The isolated BSTO film exhibits high resistivity (>10 MOhm at room temperature and increases greatly at lower temperatures). Thus, we can establish that an interface area is responsible for metallic conductivity and negative photoconductivity. Yet, the observed processes are on a timescale of seconds, therefore, these charge dynamics must be associated with relaxations in the ferroelectric film. Indeed, any relaxation processes in the region with metallic behavior are much faster. After photoexcitation, the carriers in the ferroelectric film move in the internal electric field of the film, leading to the additional screening of the polarization of the ferroelectric film [43,44,45]. This reduction in the polarization in the ferroelectric film influences the highly conductive interface area and could be responsible for the measured increase in the resistance.

One way to test this mechanism is to measure charge changes directly. We are not able to probe the interface but we can monitor the charge dynamics at the outer surface of the top BSTO film. For this purpose, we performed an AFM study of the surface potential using the Kelvin mode of AFM. Figure 5 shows the time dependence of the surface Kelvin mode potential of the BSTO/LMO/BSTO heterostructure with periodic switching of the green light, in a similar manner as in the previous experiments. The measurements were performed at room temperature. Since the transition temperature, *T*c, of the BSTO sample is above the room temperature (*T*c about 540 K [25,26]), the measurements were performed when the BSTO sample was in the ferroelectric phase. This is reasonable because ferroelectric properties do not change significantly below *T*c. Note that the BSTO sample is always in the dielectric ferroelectric phase. The metallic conductivity is due to the interface where the charge is accumulated and the polarization is screened. This follows from the fact that if we make contact with the BSTO part which does not cover the interface area, the sample never shows any metallic conductivity. The illumination leads to a drop in surface potential and then the surface potential recovers when illumination stops. The specific switching time between the dark and the photostimulated states occurs on a timescale of 100–200 s. The longer time could be related to slower screening processes at the surface, with charge collection from the air, compared to screening at the interface. This behavior is consistent with the previous assumptions about charge dynamics. We, therefore, can explain the observed negative photoconductivity effect in BSTO/LMO/BSTO heterostructure by the partial screening of the ferroelectric polarization by photostimulated charge carriers and AFM studies confirm our assumptions.

## 4. Conclusions

The Ba_0.8_Sr_0.2_TiO_3_/LaMnO_3_/Ba_0.8_Sr_0.2_TiO_3_ heterostructures on a MgO substrate with Ba_0.8_Sr_0.2_TiO_3_ ferroelectric films were studied. We found the occurrence of a metallic character of the temperature behavior of the resistance at *T* below 175 K. This behavior is associated with an increased charge concentration at the interface which appears as a result of the discontinuity of the ferroelectric polarization at the interface between the films [13]. At these temperatures, we observed the effect of negative photoconductivity on the Ba_0.8_Sr_0.2_TiO_3_/LaMnO3 interface induced by the continuous photo illumination of the heterostructure from the surface with a ferroelectric film. This behavior is opposed to the usual photoconductivity in ordinary semiconductors, where, due to photo carriers, the conductivity increases. When combined exposure to green and infrared light was used, the effects of exposure were added and a cumulative effect was observed. The observed effect cannot be explained by the direct heating of the sample by laser pulses as it was prevented by using low fluence unfocused laser beams and is not an intrinsic feature of any of the constituent materials but a result of the charge redistribution in the layers. Since the metallic conductivity is due to ferroelectric polarization, we assume that the origin of the photo resistance effect is associated with the partial screening of ferroelectric polarization by photo-excited charge carriers. Since the properties of a ferroelectric film change weakly at temperatures below *T*_c_, the studies in the Kelvin mode of atomic force microscopy were carried out at room temperature. The measurements showed that illumination affects the surface charge concentration. This evidence of the charge redistribution confirms our hypothesis.

## Figures and Tables

**Figure 1 nanomaterials-12-03774-f001:**
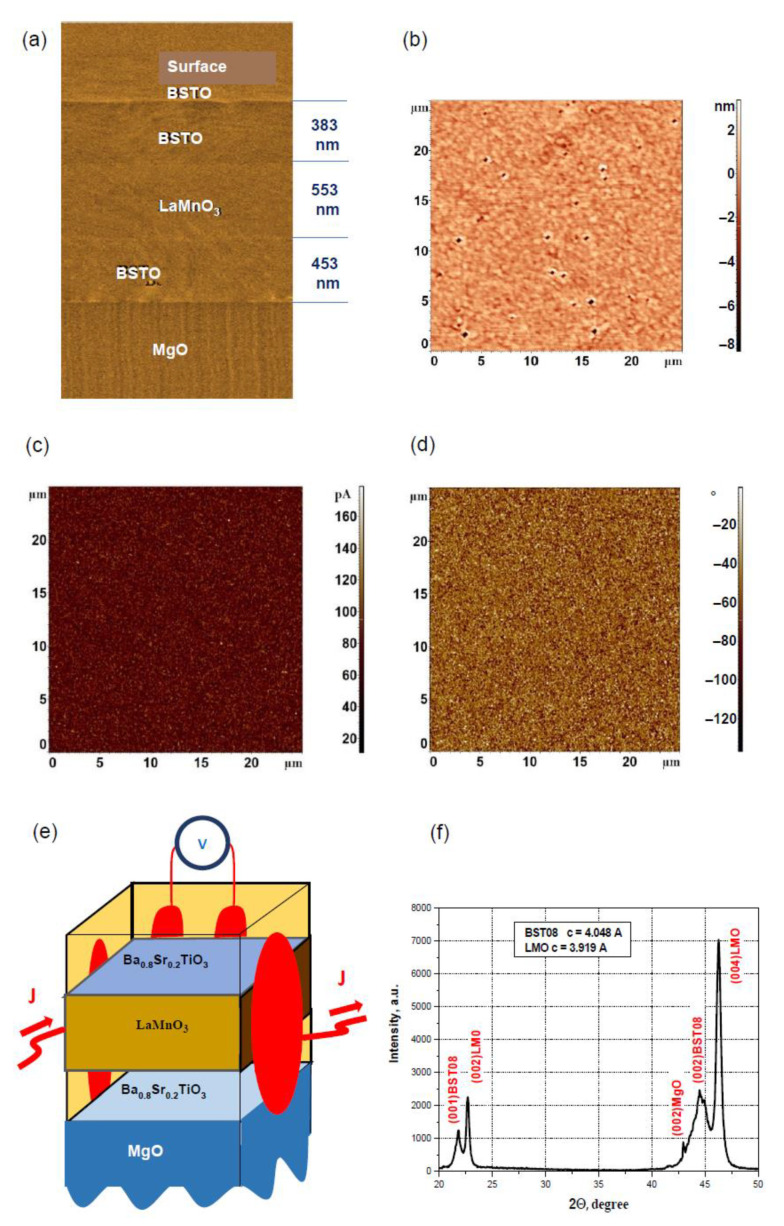
(**a**) STEM image of the BSTO/LMO/BSTO heterostructure; (**b**) the typical topography of the surface of the BSTO film of a virgin heterostructure measured by AFM with the average roughness 0.54 nm; (**c**) the amplitude and (**d**) phase of the vertical component of piezoelectric response on the surface of the BSTO film measured by AFM; (**e**) the scheme of the electrical resistance measurements with the red color schematically showing the placement of the contacts with silver paste and the current and measuring wires; and (**f**) the X-ray diffraction pattern of the heterostructure.

**Figure 2 nanomaterials-12-03774-f002:**
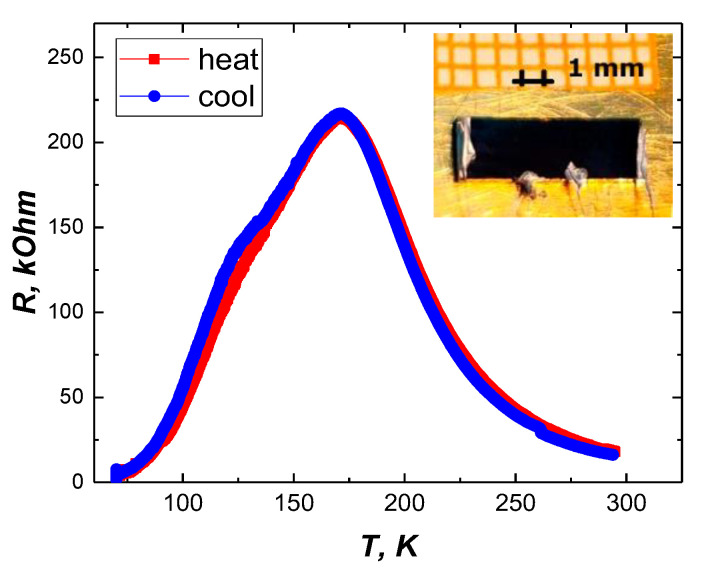
Temperature dependences of the resistance *R*(T) for the Ba_0.8_Sr_0.2_TiO_3_/LaMnO_3_/Ba_0.8_Sr_0.2_TiO_3_ heterostructure. A picture of the sample with applied electrodes is shown in the insert.

**Figure 3 nanomaterials-12-03774-f003:**
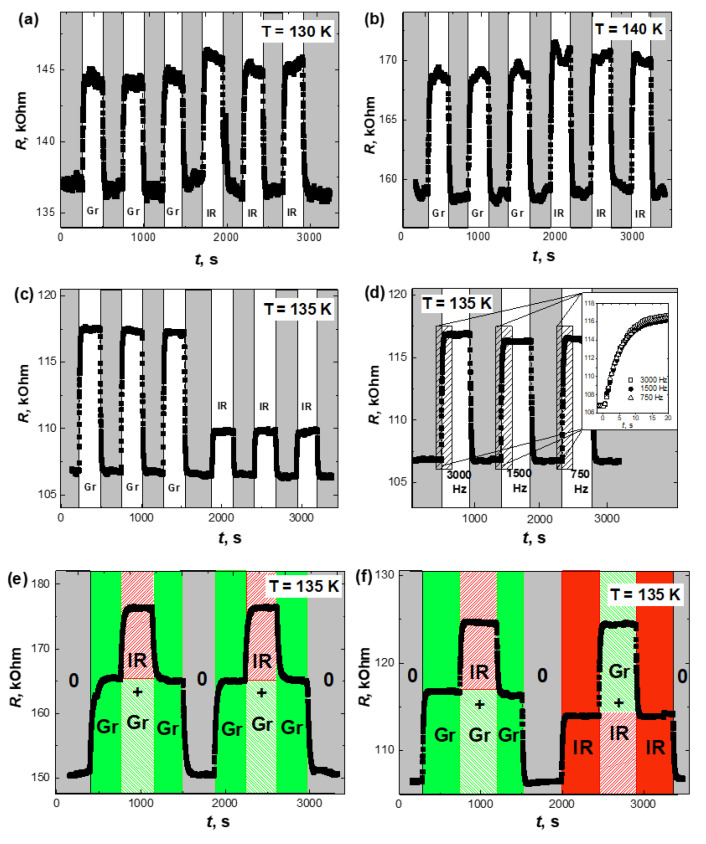
(**a**,**b**) The multiple on-off switching of electrical resistance of the Ba_0.8_Sr_0.2_TiO_3_/LaMnO_3_ heterostructure at 130 K (**a**) and 140 K (**b**) by the green (wavelength 514 nm) and infrared (wavelength 1028 nm) illumination of the same intensity (80 mW/cm^2^), starting with the dark state. (**c**) The multiple on-off switching of electrical resistance of the Ba_0.8_Sr_0.2_TiO_3_/LaMnO_3_ heterostructure at 135 K by green (80 mW/cm^2^) and infrared illumination (40 mW/cm^2^), starting with the dark state. Intensities were scaled to ensure the same flow rate of photons hitting the sample. (**d**) The multiple on-off switching of electrical resistance of the Ba_0.8_Sr_0.2_TiO_3_/LaMnO_3_ heterostructure at 135 K by a green (80 mW/cm^2^) light with a pulse repetition rate of 3 kHz, 1.5 kHz, and 0.75 kHz (starting with the dark state); inset zooms in on the initial period after turning on the light. (**e**,**f**) Dynamics of the changes in the electrical resistance of a BSTO/LMO/BSTO sample during several superimposed on/off cycles of green (wavelength 514 nm) and infrared (wavelength 1028 nm) radiation at different subsequences at 135 K. The periods of light exposure are marked: 0—dark, Gr—green light, Ir—IR light, and Gr + Ir—simultaneous exposure to green and infrared light (shown in color in the online version).

**Figure 4 nanomaterials-12-03774-f004:**
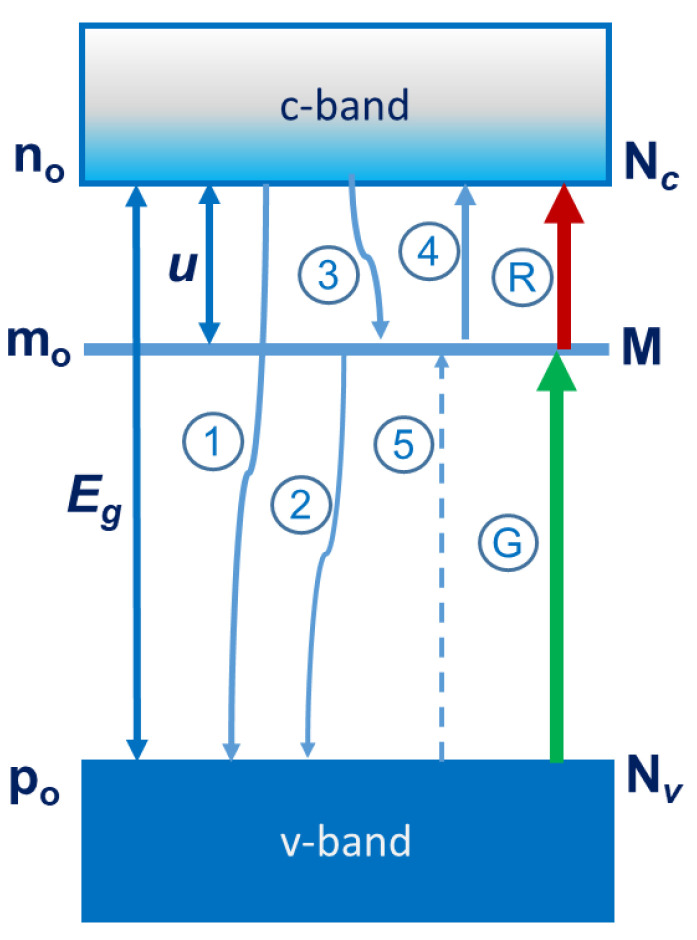
The BSTO/LMO/BSTO simplified energy diagram for the BSTO layout. C—conduction band, V—valence band, and M—trapping levels. Schematically shown are the processes of relaxation (1,2,3), green (G) and infrared (R) light irradiation, and thermal activation (4,5).

**Figure 5 nanomaterials-12-03774-f005:**
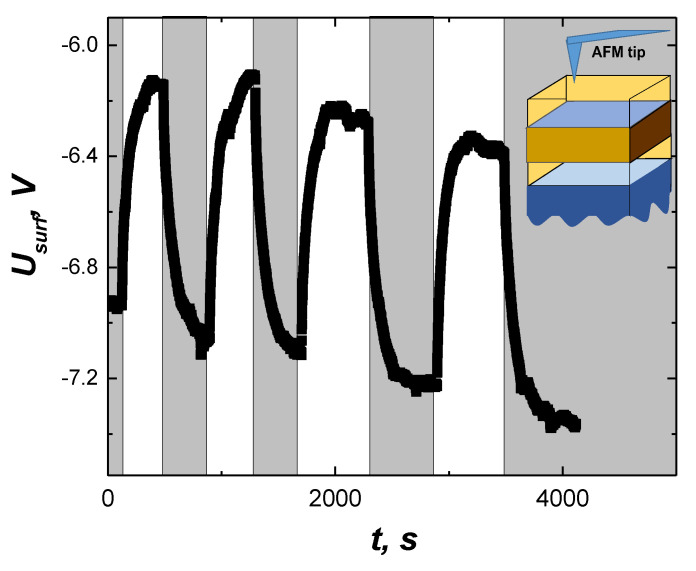
The dynamics of the surface potential of the heterostructure during several on/off cycles of green illumination (80 mW/cm^2^). An illustration of the AFM-based Kelvin probe surface potential measurements is shown in the insert.

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
