# Peer review of "Origin of Negative Photoconductivity at the Interface of Ba0.8Sr0.2TiO3/LaMnO3/Ba0.8Sr0.2TiO3 Heterostructures"

_nanomaterials, 2022, doi:10.3390/nano12213774_

Round 1

Reviewer 1 Report

The mentioned manuscript is subject to corrections.

1. The Abstract is too short. The Abstract mainly contains an enumeration of methods, but there is no general information on the results achieved. The abstract should summarize the findings of the work.

2. Literature review needs to include several recent, relevant publications (high impact) highlighting their key findings. The current version only discussed general aspects while the review of each from several papers is necessary. You may provide a review summary table consisting of a column for the comments or key conclusions.

3. Enhance the objective and novelty of the work in the introduction section.

4. Section 1. To improve the manuscript quality for photoconductivity, optical materials, and related study, cite the followings: DOIs: 10.1021/acsaelm.1c00682; 10.1021/acsaelm.1c00703; 10.1021/acsaelm.2c00069.

5. Correlate different sections' results with proper discussion.

6. The conclusion is too short. In the Conclusion section, state the most important outcome of your work. Do not simply summarize the points already made in the body — instead, interpret your findings at a higher level of abstraction. Show whether, or to what extent, you have succeeded in addressing the need stated in the Introduction.

7. Some refs are out of date (more than 10 years). Update!

8. Increase the number of recent references, especially in the introduction, and results & discussion section.

Author Response

Responses to Referee 1 and a brief summary of changes

We would like to thank the Referee for his careful reading of our manuscript, very positive criticism, and a number of comments which lead to the improvement of the manuscript.  Below we provide our response to all recommendations and criticisms. We were always trying to correct the text and clarify the situation in the places commented on by both Referees. In order to improve the presentation, we made some additional changes to the manuscript as well.  

  1. The Abstract is too short. The Abstract mainly contains an enumeration of methods, but there is no general information on the results achieved. The abstract should summarize the findings of the work.

The Abstract section was expanded to represent the major results of the work.

  1. Literature review needs to include several recent, relevant publications (high impact) highlighting their key findings. The current version only discussed general aspects while the review of each from several papers is necessary. You may provide a review summary table consisting of a column for the comments or key conclusions.

The literature review was expanded, and we make sure to include more recent works on relevant aspects of our paper.

  1. Enhance the objective and novelty of the work in the introduction section.

The Introduction section was extended to highlight the objective and novelty of our approach.

  1. Section 1. To improve the manuscript quality for photoconductivity, optical materials, and related study, cite the followings: DOIs: 10.1021/acsaelm.1c00682; 10.1021/acsaelm.1c00703; 10.1021/acsaelm.2c00069.

The suggested citations were included.

  1. Correlate different sections' results with proper discussion.

The Results section was rewritten to address the issue.

  1. The conclusion is too short. In the Conclusion section, state the most important outcome of your work. Do not simply summarize the points already made in the body — instead, interpret your findings at a higher level of abstraction. Show whether, or to what extent, you have succeeded in addressing the need stated in the Introduction.

Both the Conclusion and Introduction sections were rewritten to represent key objectives and findings.

  1. Some refs are out of date (more than 10 years). Update!

References to recent works were added. Still, some important older refs are essential.

  1. Increase the number of recent references, especially in the introduction, and results & discussion section.

References to recent works were added to Introduction and Discussion sections.

Reviewer 2 Report

please see the attached doc. 

Author Response

Responses to Referee 2 and a brief summary of changes

We would like to thank the Referees for their careful reading of our manuscript, very positive criticism, and a number of comments which lead to the improvement of the manuscript.  Below we provide our response to all recommendations and criticisms. We were always trying to correct the text and clarify the situation in the places commented on by Referee. In order to improve the presentation, we made some additional changes to the manuscript.  

  1. The author discussed studies on LAO heterojunctions from line 24 to line
    29. Starting from line 30, the author started the discussion on BSTO/LCO on
    superconductivity. Above contents are not correlated with the topic of negative
    photoconductivity of Ba0.8Sr0.2TiO3/LaMnO3/Ba0.8Sr0.2TiO3
    heterostructures. The author is suggested to discuss more on the latest
    research related to negative photoconductivity and some references are
    listed as following to give us the overall background on negative
    photoconductivity area.
    Tailor, Naveen Kumar, Partha Maity, and Soumitra Satapathi. "Observation of Negative Photoconductivity in Lead-Free Cs3Bi2Br9Perovskite Single Crystal." ACS Photonics 8.8 (2021): 2473-2480.Gustafson, Jon K., et al. "Positive and negative photoconductivity in monolayer MoS2 as a function of physisorbed oxygen." The Journal of Physical Chemistry C 125.16 (2021): 8712-8718. effect transistors." Advanced Functional Materials 31.43 (2021): 2105722.

The discussion on photoconductivity was expanded with extra refs added.

  1. Line 50, The thicknesses of the layers were 383 nm, 553 nm, and 453 nm 49
    (see Fig. 1). please improve the resolution of the STEM image with clear labels.

The layers in the STEM image were labeled more clearly.

  1. Line 56, The surface potential of top 54 Ba0.8Sr0.2TiO3 ferroelectric film of
    the heterostructure was studied by atomic force micro-55 scope (AFM).
    There is no AFM image showing surface roughness in the manuscript,
    please kindly add it to confirm the good quality of BSTO thin films.

We cannot provide information on the virgin roughness of our films, since AFM measurements were carried out at a late stage of research. And thus we provided information, at the request of the Referee, about what we have now: “The average roughness of the top BSTO layer is about 7 nm from topographic measurements using an atomic force microscope (AFM), which is much smaller than the thickness of the top layer". We do not want to focus on this issue, for the following reasons. First, the roughness of the virgin samples is about 1-2 nm. Second, we investigate the properties of interfaces in order to find the effect of ferroelectric polarization on the conductivity of the interface. Unlike the LAO/STO case, the roughness of the interface in the case of interface with ferroelectric is not so important.

  1. Line 86-96.
    please add the evidence of ferroelectricity, either AFM image or electrical
    hysteresis loop to confirm the polarization exists and the direction.

The additional measurements of the piezoelectric response from the surface of the Ba0.8Sr0.2TiO3 ferroelectric film were performed by AFM-based piezoresponse mode at room temperature. The results are shown in the Figures 1c and 1d. The presence of a piezoelectric response provides evidence in favor of the existence of ferroelectric polarization in the BSTO film. The results of these measurements show that the polarization is directed predominantly in the vertical direction.

  1. Line 61, Figure 1.
    please add detailed/clear description on the diagram of the device
    structure, especially how electrodes are placed, how ferroelectric
    polarization are oriented, how electric current is flowing from/to.

Description of the diagram was expanded.

  1. Figure 3,4,5,6 are redundant evidences of photon-induced resistance
    change. They can be combined into just one Figure.

Figures 3,4,5,6 were combined into one Figure 3.

  1. Line 152 to line 168 please add the band diagrams of BSTO/LMO/BSTO heterojunction before and after illumination to explain the resistance change observed in the
    experiment. Figure 7 only gives the band diagram of one single layer and it
    is not properly labeled on which layer it is.

We show in Fig.7 the energy diagram of the BSTO layer. The energy diagram of the LMO layer is not important because illumination occurs from the BSTO layer.

  1. Line 164, Eg=3.0-3.4 eV
    please add references on the Eg of BSTO.

Ref [33] was added

  1. Line187, The measurements are performed at room temperature.
    As the author showed in previous discussion, the temperature was selected
    to be 175K so that BSTO is at metallic state. But the AFM experiment was
    done at room temperature, in which BSTO is insulator state. The author
    uses the room temperature AFM surface potential change to be the
    evidence of the charge dynamics happening in BSTO which causes negative
    photoconductivity, which is not convincing at all.

A feasibility justification for using room temperature was added to the paper body.

Since the transition temperature, Tc of the BSTO sample is above the room temperature the measurements are performed when the BSTO sample is in the ferroelectric phase. This is reasonable because ferroelectric properties do not change significantly below Tc. Note, that the BSTO sample is always in the dielectric ferroelectric phase. The metallic conductivity is due to the interface where the charge is accumulated in order to screen the polarization. This is followed from the fact that if we make the contact on the BSTO part which does not cover the interface area the device never shows any metallic conductivity.

10.Line 211, The observed effect cannot be explained by direct heating of the
sample by laser pulses, as it was prevented by using low fluence unfocused
laser beams.
Unfocused lasers would also heat the surface or heat the connection area.
And it cannot serve as a solid evidence.
The author uses Figure 5 as the evidence to show that heating is not the
root cause by pulses lasers with different repetition rate to control the total
illumination time -in plot of the beginning portion when illumination is ON. Please add it.

We made an estimation of the heating within the simple stationary model of thermal diffusion. The estimate shows that with our parameters the excess temperature DT is of the order 0.1K. Our estimates are consistent with the estimates of DT=12 K which were performed for the focused light and at much higher intensity [D. Mihailovic and J. Demsar, ACS Symp. Ser. 730, 230 (1999).] Therefore, we believe that the heating due to the laser beam is not enough to explain the effect of negative photoconductivity. The corresponding text is added to the manuscript.

References to papers containing estimates were added. The Figure in question was added (the inset in Fig. 3d).

  1. The author failed to clearly present which interface shows the negative
    photoelectricity. It is not clearly that such effect is just bulk behavior inside
    BSTO itself or from interface

The reasoning for the BSTO/LMO interface showing the negative photoconductivity was added to the paper.

BSTO/LMO interface shows the effect of metallic conductivity. This follows from the fact that if we apply the contacts on the BSTO side of the device without contact with the interface area the metallic conductivity is absent. The metallic conductivity is observed only if the contacts cover the interface area. This observation indicates that the interface area is responsible for metallic conductivity and for negative photoconductivity.

Round 2

Reviewer 1 Report

Recommend for acceptance 

Author Response

 Responses to Referee 1 and a brief summary of changes
We would like to thank the Referee for his careful reading of our manuscript, very
positive criticism, and a number of comments which lead to the improvement of the manuscript.

Reviewer 2 Report

Please see the comment below, #1 is the main concern I think.

1.       “Figures 1c and 1d show the amplitude and the phase of the vertical com ponent of piezoelectric response on the surface of BSTO film measured by AFM at room temperature. The presence of a piezoelectric response provides evidence in favor of the existence of ferroelectric polarization in the BSTO film”

No, the picture here is not confirming the piezoelectricity. Even substrate itself can give you figure c and figure d. To confirm the vertical electricity. The author has to use DC voltage to write patterns or switch piezoelectric polarization to get hysteresis loops.

Given a random example of how to characterize PFM here, in Figure 3 of below reference:
Liu, Fucai, et al. "Room-temperature ferroelectricity in CuInP2S6 ultrathin flakes." Nature communications 7.1 (2016): 1-6.

The key idea of this manuscript is partial screening of the ferroelectric polarization by photogenerated charge carriers induced resistance change. So it would be very important for the author to provide solid evidence to at least show audience on the change of ferroelectric/piezoelectric polarization.
I cannot find any proof on piezoelectricity or ferroelectricity. Figure 1c/d could be just noise signa from PFM test. The author must prove this point.  

2.       Line 78 says BSTO roughness is <10nm. But there is no AFM picture showing such in Figure 1.

For figure 1a, please add the thickness also, not only the heterojunction. 

Author Response

Responses to Referee 2 and a brief summary of changes

We would like to thank the Referees for their careful reading of our manuscript, very positive criticism, and a number of comments which lead to the improvement of the manuscript.  Below we provide our response to all recommendations and criticisms. We were always trying to correct the text and clarify the situation in the places commented on by Referee. In order to improve the presentation, we made some additional changes to the manuscript.  

Answer on Comments and Suggestions of Reviewer 2 - Round 2

  1. “Figures 1c and 1d show the amplitude and the phase of the vertical com ponent of piezoelectric response on the surface of BSTO film measured by AFM at room temperature. The presence of a piezoelectric response provides evidence in favor of the existence of ferroelectric polarization in the BSTO film”

No, the picture here is not confirming the piezoelectricity. Even substrate itself can give you figure c and figure d. To confirm the vertical electricity. The author has to use DC voltage to write patterns or switch piezoelectric polarization to get hysteresis loops.

Given a random example of how to characterize PFM here, in Figure 3 of below reference:
Liu, Fucai, et al. "Room-temperature ferroelectricity in CuInP2S6 ultrathin flakes." Nature communications 7.1 (2016): 1-6.

The key idea of this manuscript is partial screening of the ferroelectric polarization by photogenerated charge carriers induced resistance change. So it would be very important for the author to provide solid evidence to at least show audience on the change of ferroelectric/piezoelectric polarization.
I cannot find any proof on piezoelectricity or ferroelectricity. Figure 1c/d could be just noise signa from PFM test. The author must prove this point.  

We know that a “usual” piezoelectric response measurement cannot give us confirmation of the existence of a ferroelectric phase. Therefore, we wrote very softly:

“The presence of a piezoelectric response provides evidence in favor of the existence of ferroelectric polarization in the BSTO film.” – only  “… evidence in favor ....”

We also know about the possibility to confirm the ferroelectricity it is possible to apply DC voltage to rewrite domain patterns by switching piezoelectric polarization to get hysteresis loops using piezoelectric response mode of AFM (like in Ref. [." Nature communications 7.1 (2016): 1-6.]). But in Ref. [." Nature communications 7.1 (2016): 1-6.] an electric field in the sample of about 100 kV/cm (4V/400nm) or higher was used to reverse the polarization.

In our case, it is impossible without the bottom electrode under the film, which is absent. If using a bottom electrode under the MgO substrate all voltage (10V for AFM) will drop in the substrate, because the substrate thickness is much larger than the film thickness and, in addition since the dielectric constant of the film is much larger than that in the substrate.

In our case, the results of X-ray measurements indicate that the BSTO film is in the tetragonal phase, and hence, in the ferroelectric state. Now the X-ray diffraction pattern of the heterostructure is shown in Fig. 1f.

  1. Line 78 says BSTO roughness is <10nm. But there is no AFM picture showing such in Figure 1.

For figure 1a, please add the thickness also, not only the heterojunction. 

To confirm the good quality of our BSTO thin films it is important to

know the properties of virgin samples. Therefore, we show the typical

topography of the surface of the BSTO film of the virgin heterostructure

in Fig. 1b. The average roughness for this case is 0,54 nm.

We added the value of the thicknesses in Figure 1a.
